# Duality of Branched-Chain Amino Acids in Chronic Cardiovascular Disease: Potential Biomarkers versus Active Pathophysiological Promoters

**DOI:** 10.3390/nu16121972

**Published:** 2024-06-20

**Authors:** Daniela Maria Tanase, Emilia Valasciuc, Claudia Florida Costea, Dragos Viorel Scripcariu, Anca Ouatu, Loredana Liliana Hurjui, Claudia Cristina Tarniceriu, Diana Elena Floria, Manuela Ciocoiu, Livia Genoveva Baroi, Mariana Floria

**Affiliations:** 1Department of Internal Medicine, “Grigore T. Popa” University of Medicine and Pharmacy, 700115 Iasi, Romania; tanasedm@gmail.com (D.M.T.); ank_mihailescu@yahoo.com (A.O.); diana-elena.iov@d.umfiasi.ro (D.E.F.); floria_mariana@yahoo.com (M.F.); 2Internal Medicine Clinic, “St. Spiridon” County Clinical Emergency Hospital, Iasi 700111, Romania; 3Department of Ophthalmology, “Grigore T. Popa” University of Medicine and Pharmacy, 700115 Iasi, Romania; costea10@yahoo.com; 42nd Ophthalmology Clinic, “Prof. Dr. Nicolae Oblu” Emergency Clinical Hospital, 700309 Iași, Romania; 5Department of General Surgery, Faculty of Medicine, “Grigore T. Popa” University of Medicine and Pharmacy, 700115 Iasi, Romania; dscripcariu@gmail.com; 6Regional Institute of Oncology, 700483 Iasi, Romania; 7Department of Morpho-Functional Sciences II, Physiology Discipline, “Grigore T. Popa” University of Medicine and Pharmacy, 700115 Iasi, Romania; loredana.hurjui@umfiasi.ro; 8Hematology Laboratory, “St. Spiridon” County Clinical Emergency Hospital, 700111 Iasi, Romania; 9Department of Morpho-Functional Sciences I, Discipline of Anatomy, “Grigore T. Popa” University of Medicine and Pharmacy, 700115 Iasi, Romania; cristinaghib@yahoo.com; 10Hematology Clinic, “Sf. Spiridon” County Clinical Emergency Hospital, 700111 Iasi, Romania; 11Institute of Gastroenterology and Hepatology, “St. Spiridon” County Clinical Emergency Hospital, 700111 Iasi, Romania; 12Department of Pathophysiology, Faculty of Medicine, “Grigore T. Popa” University of Medicine and Pharmacy, 700115 Iasi, Romania; mciocoiu2003@yahoo.com; 13Department of Surgery, Faculty of Medicine, “Grigore T. Popa” University of Medicine and Pharmacy, 700115 Iasi, Romania; livia.baroi@umfiasi.ro; 14Department of Vascular Surgery, “St. Spiridon” County Clinical Emergency Hospital, 700111 Iasi, Romania

**Keywords:** branched-chain amino acids, cardiovascular diseases, heart failure, type 2 diabetes, insulin resistance, mammalian target of rapamycin (mTOR)

## Abstract

Branched-chain amino acids (BCAAs), comprising leucine (Leu), isoleucine (Ile), and valine (Val), are essential nutrients vital for protein synthesis and metabolic regulation via specialized signaling networks. Their association with cardiovascular diseases (CVDs) has become a focal point of scientific debate, with emerging evidence suggesting both beneficial and detrimental roles. This review aims to dissect the multifaceted relationship between BCAAs and cardiovascular health, exploring the molecular mechanisms and clinical implications. Elevated BCAA levels have also been linked to insulin resistance (IR), type 2 diabetes mellitus (T2DM), inflammation, and dyslipidemia, which are well-established risk factors for CVD. Central to these processes are key pathways such as mammalian target of rapamycin (mTOR) signaling, nuclear factor kappa-light-chain-enhancer of activate B cells (NF-κB)-mediated inflammation, and oxidative stress. Additionally, the interplay between BCAA metabolism and gut microbiota, particularly the production of metabolites like trimethylamine-N-oxide (TMAO), adds another layer of complexity. Contrarily, some studies propose that BCAAs may have cardioprotective effects under certain conditions, contributing to muscle maintenance and metabolic health. This review critically evaluates the evidence, addressing the biological basis and signal transduction mechanism, and also discusses the potential for BCAAs to act as biomarkers versus active mediators of cardiovascular pathology. By presenting a balanced analysis, this review seeks to clarify the contentious roles of BCAAs in CVD, providing a foundation for future research and therapeutic strategies required because of the rising prevalence, incidence, and total burden of CVDs.

## 1. Introduction

Branched-chain amino acids (BCAAs), including leucine, isoleucine, and valine, are essential nutrients that play critical roles in protein synthesis, energy production, and metabolic regulation. While BCAAs are essential for normal physiological processes, emerging research has highlighted their complex involvement in chronic cardiovascular diseases (CVDs), including heart failure (HF), atherosclerosis (AS), coronary artery disease (CAD), and hypertension [1]. The function of BCAAs in chronic CVD is multifaceted and involves intricate interactions with various metabolic pathways, cellular signaling, and mechanisms [2].

Alterations in metabolism and circulating BCAA levels have been observed and are increasingly recognized as potential biomarkers, and they are associated with insulin resistance, dyslipidemia, and endothelial dysfunction [3,4]. Moreover, the role of BCAAs in chronic CVD extends beyond their traditional functions in protein metabolism. BCAAs can modulate cellular signaling pathways, such as the mammalian target of rapamycin (mTOR) pathway, which plays a crucial role in regulating cell growth, proliferation, and metabolism [5]. Dysregulated mTOR signaling, influenced by BCAA availability, has been implicated in adverse cardiac remodeling, fibrosis, and hypertrophy observed in heart failure, as well as other cardiovascular conditions [5].

Furthermore, the interaction between BCAAs and lipid metabolism, particularly through sterol regulatory element-binding proteins (SREBPs) and peroxisome proliferator-activated receptors (PPARs), underscores the intricate metabolic cross-talk in CVD pathogenesis [6,7]. Dysregulation of these transcriptional regulators can lead to lipid accumulation, inflammation, and oxidative stress, all of which contribute to the development and progression of atherosclerosis and coronary artery disease. Besides their role in CVD pathology, recent studies have correlated BCAAs with type 2 diabetes (T2D) [6,7], not only showing their involvement in glycemic control but also their involvement in insulin resistance [3,4].

In this context, the involvement of BCAAs in chronic CVD represents a dynamic interplay between nutrient metabolism, cellular signaling, and disease pathophysiology. Understanding the nuanced roles of BCAAs in cardiovascular health and disease holds promise for identifying novel therapeutic targets and interventions aimed at mitigating metabolic dysfunction and improving outcomes for individuals with chronic cardiovascular conditions. This review aims to explore and elucidate the multifaceted interplay of BCAAs in chronic CVD, highlighting the latest studies about their impact on metabolic regulation, cellular signaling, and disease pathogenesis. By dissecting these complex interactions, we can advance our understanding of CVD mechanisms and identify new avenues for personalized treatment strategies targeting BCAA metabolism and associated pathways.

## 2. Branched-Chain Amino Acid Synthesis, Metabolism, and Catabolites

BCAAs, specifically leucine, isoleucine, and valine, have gained interest since the mid-1970s and 1980s due to their observed immunomodulatory properties subsequent to their broad implementation as dietary supplements in individuals affected by chronic liver and kidney disease, sepsis or trauma, and muscle-wasting disorders [2,8,9]. As stimulators of the mTOR pathway, notably so in the case of leucine, BCAAs indirectly interfere with numerous other signaling pathways that mediate insulin action, crucial for maintaining glucose homeostasis, protein synthesis, mitochondrial biogenesis, inflammation, and lipid metabolism [10,11]. Despite the numerous in vitro research studies conducted on cellular cultures and human and animal model studies, the results have been contradictory regarding the biological activity of BCAAs [8,12]. Consequently, the intricate mechanisms underlying their effects still need to be more adequately understood [8,12].

Although they constitute about 35% of essential amino acids in mammals, playing a significant role in protein structure, BCAAs cannot be synthesized and must be obtained from dietary sources, representing approximately 20–25% of daily dietary protein intake [13,14]. In order to regulate catabolism in mammals, a tightly controlled metabolic system has evolved, consisting of a cascade of processes that include transamination, irreversible oxidative decarboxylation, and adenosine triphosphate (ATP) generation [15,16].

### 2.1. Transamination

The initial event in the breakdown of the three aforementioned BCAAs is multifaceted. It involves the conversion of the amino acids to branched-chain α-keto acids (BCKAs) through transamination, a process catalyzed by the cytosolic or mitochondrial enzyme BCAA aminotransferase (BCAT) [17]. The transamination process involves the transfer of specific amino groups to α-ketoglutarate (α-KG), resulting in glutamate and the formation of corresponding BCKAs. The BCKAs derived from leucine, isoleucine, and valine are α-ketoisocaproate (α-KIC), α-keto-β-methylvalerate (α-KMV), and α-ketoisovalerate (α-KIV), respectively [18,19,20,21]. Further, glutamate can be used either as the amino component necessary to form alanine (ALA) or it can be used for ammonia detoxification, resulting in glutamine (GLN), and all of the above-mentioned three components involved in transamination (BCKAs, ALA, and GLN) are released into the bloodstream via muscles [18,19,20,21].

The entire transamination process cannot occur without the action of the BCAT enzyme, along with its two isoforms. BCAT1, the cytosolic one, is predominantly expressed in the brain, while BCAT2, the mitochondrial one, is found in most tissues and particularly skeletal muscle [14,21,22,23]. Due to the reduced activity of BCAT in the liver as opposed to other amino acids, the first step in the catabolism of BCAAs does not occur at this level, and they rapidly reach elevated levels in the systemic circulation after protein ingestion and are even more quickly available to extrahepatic tissues [22]. Transamination is subsequently followed by the decarboxylation and dehydrogenation of the BCKAs to branched-chain acyl-CoA esters, along with the production of CO2 and NADH by the BCKA dehydrogenase complex (BCKDC/BCKDH). This step converts α-KIC, α-KMV, and α-KIV to isovaleryl-CoA, 2-methyl-butyryl-CoA, and isobutyryl-CoA, respectively [14,20,24].

### 2.2. Decarboxylation

Following the transamination pattern described above, the catalyst of the oxidative decarboxylation reaction is the BCKDH complex, which is a multienzyme complex located in the inner mitochondrial membrane [25,26]. The complex consists of three catalytic components: E1 (thiamine-dependent decarboxylase) for the oxidative decarboxylation of BCKAs, E2 (dihydrolipoyltransacylase) for transferring acyl groups to CoA, and E3 (FAD-dependent dihydrolipoyl dehydrogenase) for electron transfer to NAD^+^ [25,26].

E1 uses a reduced coenzyme-A substrate for decarboxylation, while E2 uses lipoic acid as an acceptor and transfers it to acetyl-CoA [13,27]. The phosphorylation status of E1 is changed by the different levels of BCKDH kinase (BCKDK) and protein phosphatase Mg2^+^/Mn2^+^-dependent 1K (Ppm1k), which then regulates the inhibitory phosphorylation of BCKDH E2 and impacts the flow of substances in the pathway [13,24,25]. E3 acts as a lipoamide dehydrogenase, transferring its hydrogen to NAD^+^ via FAD [13,27].

The phosphorylation–dephosphorylation mechanism regulates the BCKDH complex, with functional alterations in kinase activity producing a significant impact. BCKDK causes inactivation through phosphorylation, while mitochondrial phosphatase 2C (PP2Cm) catalyzes dephosphorylation, which activates the BCKDH complex [17,28]. The highest BCKDK activity is exhibited by the liver, followed by the kidneys and heart, while the muscles, adipose tissue, and brain have lower activity levels [13,23].

### 2.3. ATP Generation

Following the decarboxylation of BCKAs induced by the BCKDH complex, the subsequent stages of degradation resemble the process of fatty acid oxidation in terms of adenosine triphosphate (ATP) production [29]. However, each reaction is distinct from the three BCAAs, and the sole location in which these reactions to occur is the mitochondrial matrix. The end result of valine is propionyl-CoA, which eventually enters the tricarboxylic acid (TCA) cycle and has the potential to be used in gluconeogenesis [29]. On the other hand, leucine produces acetyl-CoA, making it a ketogenic source. Isoleucine, however, produces both propionyl-CoA and acetyl-CoA [8,23,30,31].

Even though their metabolism involves multiple tissues, the levels of BCAAs in systemic circulation vary depending on dietary intake and general protein turnover [32]. Through their metabolic particularity of not being broken down in the liver as other amino acids, BCAAs are absorbed at the intestinal level and enter the bloodstream directly; therefore, the levels of BCAAs in the diet directly impact the levels in the blood [16,33]. Excess amino acids are stored in proteins in the liver and skeletal muscle, acting as major reservoirs for amino acids in mammals, and they are utilized during fasting periods [8,31]. After being released from proteins, BCAAs are metabolized in various peripheral tissues that express BCAT enzymes, such as skeletal muscle, as described above [8,31]. This leads to the generation of BCKAs, which are released into the bloodstream. Although the liver lacks BCAT expression, it plays a role in the metabolism of circulating BCKAs, using their carbon group for gluconeogenesis, ketogenesis, or fatty acid synthesis via the BCKDH complex [23,34].

Isotopic tracer studies that examined the uptake and breakdown of BCAAs in the whole body sought to uncover tissue-specific differences in BCAA metabolism and maximum oxidation occurring in skeletal muscle [25]. In addition, skeletal muscle tissues such as the liver and pancreas contribute the most to protein synthesis from BCAAs, and metabolic alteration in these tissues, as seen in neoplasms, may be responsible for consistent alterations in BCAA levels [25].

## 3. Branched-Chain Amino Acid-Regulated Signaling Pathways

Furthermore, apart from their roles as structural and metabolic components, BCAAs also have a functional role, acting as a promoter of various cellular signaling pathways. Among these pathways, the most extensively studied is the activation of the mTOR signaling pathway by leucine [26,35]. mTOR, a serine–threonine protein kinase, is a member of the PI3K-related kinase (PIKK) family and functions as the catalytic component for two separate complexes known as mTORC1 and mTORC2 [36,37,38]. Besides controlling the production of proteins by overseeing the messenger ribonucleic acid (mRNA) translation process, the intracellular mTOR molecule serves as a critical regulator in various essential cellular processes, including cell growth, cell division, autophagy, and glucose balance [39].

### 3.1. mTOR and Upstream Regulators

mTORC1 is composed of five elements: mTOR, the regulatory protein linked to mTOR (Raptor); the G-protein beta-subunit-like protein (GbL or mammalian lethal with Sec13 protein 8—mLST8); and two inhibitory components, pleckstrin (DEP) domain-containing mTOR-interacting protein (Deptor, Egl-10) and proline-rich AKT substrate 40 kDa (PRAS40) (Figure 1) [10,16,35,39]. Conversely, mTORC2 consists of mTOR, Deptor, mLST8, Rictor (rapamycin-insensitive companion of mTOR), protein observed with Rictor-1 (Protor-1), and mammalian stress-activated protein kinase-interacting protein 1 (mSIN1) [10,16,35,39].

Upon ingestion, BCAAs are transported into cells and undergo metabolism, primarily in skeletal muscle. The first step involves transamination, where BCAAs are converted into BCKAs by the enzyme BCAT. Subsequently, the BCKAs are further metabolized in mitochondria through a series of enzymatic reactions, ultimately generating intermediates like acetyl-CoA and succinyl-CoA, which enter central metabolic pathways [10,16,35,39]. The mTOR pathway is particularly sensitive to changes in BCAA levels, especially leucine. Leucine acts as a direct activator of mTORC1 (mTOR complex 1) by binding to and activating the protein kinase mTOR [11,26,32]. The activation of mTORC1 stimulates protein synthesis by phosphorylating downstream effectors like S6 kinase 1 (S6K1) and eukaryotic translation initiation factor 4E-binding protein 1 (4E-BP1), leading to the enhanced translation of mRNA into proteins involved in cell growth and proliferation [11,26,32].

The activation of the mTORC1 signaling pathway is initiated by a variety of factors, including cytokines, hormones, growth factors, and nutrients, further leading to the phosphorylation of translational regulators, including S6K, 4E-BP1 (eukaryotic translation initiation factor 4E (eIF4E)-binding protein 1), and SREBP1 [11,26,32]. This gamut of pathways further oversees the regulation of new lipid production, aerobic glycolysis, and the pentose phosphate metabolic pathway, also bringing equity among cell growth commensurate with nutrient availability [11,26,32]. Conversely, mTORC2 contributes to cell survival, cell proliferation, and the regulation of the cytoskeleton, being indirectly subjected to the control of the mTORC1 signaling pathway [11,13].

The tuberous sclerosis complex (TSC), composed of TSC1 (hamartin) and TSC2 (tuberin), is the primary mechanism for the upstream regulation of mTORC1 [40]. The stimulation of mTORC1 by growth factors like insulin and insulin-like growth factor 1 (IGF-1) triggers the activation of the phosphatidylinositol 3-kinase (PI3K)/protein kinase B (Akt) complex and extracellular-signal-regulated kinase 1 and 2 (ERK1/ERK2) [40]. These, in turn, phosphorylate and inhibit the activity of the TSC1–TSC2 complex, leading to the formation of a suppressor complex that acts as a guanine triphosphatase (GTPase)-activating protein (GAP) for the Ras homolog enriched in brain (Rheb), activating mTORC1 [40,41,42]. On the contrary, the AMP-activated protein kinase (AMPK) pathway activates TSC1/2, resulting in the inactivation of Rhed and mTORC1 inhibition with energy restriction. In particular, the AMPK pathway acts as an inhibitor, regardless of the TSC1/2 complex, due to direct Raptor phosphorylation [40,41]

Another pathway of mTORC1 activation that acts independently of TSC1/2 runs through Rag GTPases due to elevated levels of amino acids, particularly leucine. A complex between Rag GTPases and mTORC1 is formed in the presence of amino acids translocating from the cytoplasm to the lysosomal membranes to further activate by Rheb [43]. The direct interaction with the Raptor part of mTORC1 is facilitated by leucyl-tRNA synthetase (LRS)’s interaction with Rag, resulting in the assembly of the RagA/B–RagC/D complex in their GTP/GDP-loading states (RagA/B-GTP and RagC/D-GDP) [39,44]. This explains why low levels or the absence of amino acids promote mTORC1 inactivation secondary to Rag transitions into an inactive state [45]. This mTORC1-independent pathway activation, being primarily impacted by leucine levels compared to other amino acids, justifies why the reduction in leucine alone is just as effective as completely depriving the body of all amino acids and, conversely, why leucine alone is enough to facilitate the activation [14].

Two GAP complexes, GATOR1 and Folliculin (FLCN)–folliculin interacting protein 2 (FNIP2), are responsible for the downregulation of the Rag GTPases. GATOR1 acts on RagA and RagB, while FLCN-FNIP2 acts on RagC and RagD [14,44,46]. The GATOR2 complex, composed of Sestrin2 and other interacting proteins, negatively regulates GATOR1 [14,44,46]. Sestrin2, being a leucine sensor upstream of mTORC1, acts as a negative regulator, whereas LRS acts as a positive regulator of mTORC [14,44,46].

Although the mechanisms of mTORC1 activation are extensively studied, the mechanisms of mTORC2 activation still need to be better understood [14]. Recently issued hypotheses posit that its activation is induced by growth factors through the binding of mSin1 in response to PI3K being stimulated by growth factors like insulin [14]. When stimulated by amino acids, mTORC2 is involved in various cellular functions, such as adhesion, differentiation, proliferation, and apoptosis, functions mediated by the activation of Akt (Serine473), protein kinase C (PKC) alpha, and serum- and glucocorticoid-inducible kinase (SGK) [47].

### 3.2. Downstream Regulators

In addition to the above-described upstream regulators of mTORC, the downstream regulators appear to have greater involvement in insulin resistance [48].

The phosphorylation of the eukaryotic initiation factor (eIF) 4E-binding protein 1 (4E-BP1), an inhibitory protein, enhances the formation of the elF4F complex, which promotes mRNA translation [11,26,32]. Moreover, the activation of ribosomal S6K, also facilitated by insulin and amino acids, provides an additional trigger for translation initiation [32]. The simultaneous activation of mTOR and S6K1 negatively affects Akt by phosphorylating serine residues on IRS-1, disrupting the interaction between insulin and its receptor [2,47]. Consistently elevated BCAA and mTORC1 activation levels trigger a negative feedback loop of IRS-1, resulting in insulin resistance [2,47].

The impact of mTORC2 on metabolism, stress responses, and apoptosis is also mediated by phosphorylating several AGC kinases, such as Akt, serum- and glucocorticoid-induced protein kinase (SGK), protein kinase C-a (PKCa), and Rho1 GDP-GTP exchange protein-2 (Rom2), with Akt subsequently inhibiting TSC1/2 [49]. Furthermore, mTORC2 promotes the negative feedback of IRS-1 and prevents its inactive form from cytosol buildup [49].

PPARs, with their three subtypes (PPAR α, β/δ, γ), are nuclear receptors that play a significant role in regulating the metabolism of BCAAs in relation to mTOR in insulin resistance. The interaction between PPARs and BCAAs is complex and can impact various aspects of metabolic homeostasis [50,51,52]. PPARα is primarily involved in regulating fatty acid metabolism and energy homeostasis, being highly expressed in tissues such as the heart, liver, and skeletal muscle. The activation of PPARα promotes the uptake of fatty acids into cardiomyocytes and enhances mitochondrial fatty acid oxidation [50,51,52]. Studies have suggested that PPARα activation may indirectly influence BCAA metabolism through its effects on mitochondrial function and energy utilization, therefore increasing the risk of lipid peroxidation toxicity and the susceptibility of the heart to ischemia/reperfusion damage via the GCN2/ATF6 pathway [53]. PPARδ is predominantly expressed in adipose tissue and plays a key role in adipocyte differentiation and insulin sensitivity. While less studied in the context of BCAA metabolism compared to PPARα, in one study, PPARδ inhibition modulated cardiomyocyte proliferation, while the overexpression via the agonist GW5015116 reduced the proliferation following ischemic lesions and significantly reduced BCAA and secondary metabolite levels, alleviating insulin resistance in the diabetic milieu [50,51].

Significantly, these modifications replicate the first metabolic shifts associated with systemic insulin resistance [52]. It is unclear whether elevated levels of fatty acid oxidation are preserved or reduced in subsequent stages of diabetic cardiomyopathy, as happens in the ischemic heart [52].

## 4. Branched-Chain Amino Acid in Cardiovascular Disease

Being one of the vital organs requiring major energy consumption on a daily basis, when subjected to pathological energetic states such as pressure overload, hypertrophy, and ischemia, a metabolic shift must occur to enable adaptation to different physiological and nutritional circumstances [54].

The majority of the energy is normally produced through fatty acid oxidation in the mitochondria, with a return to the fetal pattern of cardiac development in a pathological state, in which anaerobic glycolytic metabolism, with its high use of ketones and lactate and a decrease in fatty acid use, is characteristic [50,55]. This alternative adaptive mechanism is only beneficial in the short-term, extended use of glucose, causing a deficiency in ATP production commonly observed in various cardiac diseases [37,56,57,58]. ATP generation through the tricarboxylic acid (TCA) cycle in the mitochondria after fatty acid breakdown, together with the electrons generated and transferred through the electron transport chain (ETC), leads to the creation of a proton gradient and energy production [50,59,60]. This explains why fatty acids are the primary source of energy for the adult heart and why glucose is a secondary energy source [50,59,60].

The cardiac muscle is only a minor site of BCAA oxidation in terms of mass compared with other tissues like skeletal muscle and fat. Therefore, impairments in non-cardiac BCAA oxidation are more likely to be responsible for the increased levels of circulating BCAAs in CVD [60,61]. A potential association could be explained by a decrease in the breakdown of BCAAs due to obesity or insulin resistance cases in which the metabolism exhibits inflexible alterations characterized by a resistance to shift from oxidizing fatty acids to oxidizing glucose, subsequently leading to an increase in triglyceride levels, particularly in the heart tissue [26,62]. Dysfunctional BCAA breakdown and a deficiency in the enzymes responsible for their metabolism lead to increased levels in both the heart tissue and liver, with the subsequent mTORC-1 activation leading to cardiac hypertrophy [16,63]. Additional proof that alterations in the metabolism of BCAAs are connected to CVD was observed in the Prevención con Dieta Mediterránea (PREDIMED) study, a case–cohort study involving 226 newly diagnosed CVD cases and 781 individuals without CVD which concluded that elevated levels of BCAAs were linked to a higher risk of CVDs, particularly stroke (Table 1) [64]. However, the significance of BCAA metabolism in relation to CVD is still not well understood [65].

Understanding the mechanisms involved in the pathogenesis of CVD and the role of BCAA may provide valuable insights for developing new strategies, especially when they coexist with type 2 diabetes mellitus (T2DM) encompassing the cardiometabolic disease [34,79,80,81].

### 4.1. Heart Failure

A decrease in energy production from the breakdown of fatty acids at the expense of increased glucose utilization marks the defining metabolic substrate for heart failure. This metabolic inflexibility contributes to the progression of both systolic and diastolic dysfunctions [50,82].

In order to gain a deeper understanding of the role of amino acid metabolism in heart failure, particular attention has been given to BCAAs and their different utilization patterns [50,82]. The changes observed in failing hearts mainly encompass the accumulation of BCAAs secondary to multiple probable mechanisms, such as the downregulation of encoding genes from the catabolic pathway, mitochondrial dysfunction, mTOR activation, and glucose oxidation impairment [36,60,83,84].

Several animal models of heart failure were studied in correlation with BCAA levels, but the results were conflicting, giving scope for further research in the future. In rodents with HF induced by myocardial infarction (MI), BCAA systemic levels increase progressively, while both cardiac BCAA levels and catabolic enzyme abundance decrease secondary to a decreased metabolism over time [85,86].

mTOR activation and reactive oxygen species (ROS) lead to the accumulation of BCAAs and their catabolic enzymes, which is a hallmark of cardiac dysfunction and remodeling [87]. While BCKDH subunits are downregulated in heart failure patients, BCAAs accumulate in the myocardium, but these do not always correspond to plasma concentrations [16,60,88,89]. Consequently, therapeutic methods for promoting BCAA breakdown in myocardial tissue have been tested, but the depletion of plasma levels has also led to exacerbations of cardiac dysfunction, as seen in a murine model of left ventricular failure [63]. Another model of pressure overload-induced HF in transaortic constriction (TAC) mice resulted in increased cardiac BCAA catabolic intermediates, which was corroborated by subsequent verification from a human cohort with HF with reduced ejection fraction (HfrEF) that displayed increased levels of a-keto-b-methylvalerate (the BCKA of isoleucine) [90,91]. Another comparative study on TAC- and MI-induced HF revealed changes in cardiac BCAA levels with no alteration regarding plasmatic values but with a new insight into the correlation with cardiac insulin resistance [16,92].

In 2022, a novel ‘chemogenetic’ mouse model of dilated cardiomyopathy-induced heart failure (HF) was introduced utilizing hydrogen peroxide in cardiomyocytes [93]. After feeding animals with the d-alanine enzyme, oxidative stress was induced [93]. BCAA catabolic enzymes were downregulated with increased cardiac levels but without a change in the plasma concentrations of the BCAAs [93]. In a different study based on dilated cardiomyopathy but in human patients, higher BCAA cardiac levels outlined a deficiency in BCKDH phosphorylation and Ppm1k and BCAT2 activity [66]. These results show model-dependent differences in systemic circulating BCAA levels and point to higher cardiac BCAA concentrations and impaired cardiac BCAA metabolism as consistent hallmarks of cardiac dysfunction [58,94,95].

To differentiate the extent to which intrinsic disturbances in cardiac amino acid metabolism versus fluctuating plasma amino acid levels influence the physiopathology of HF, further studies were carried out employing dietary BCAA supplementation, pharmacological interventions targeting BCAA metabolism, and genetic animal models [17,85].

A novel mice model lacking Ppm1k (or Pp2cm), a gene encoding a protein phosphatase crucial for BCAA catabolism, exhibits heightened levels of BCAAs and related metabolites in both plasma and cardiac tissue, alongside a mild decline in cardiac function over time [28,50]. Notably, when subjected to cardiac stress via transaortic constriction (TAC), Ppm1k-deficient mice display accelerated HF progression [90]. In Ppm1k-deficient mice, the allosteric inhibitor of BDK, known as 3,6 dichlorobenzo[b]thiophene-2-carboxylic acid (BT2), was used in an attempt to uncover the cardiac-specific metabolic BCAA dysfunction [56]. The administration of BT2 for one week resulted in the dephosphorylation and maximal activation of BCKDH in multiple sites, including the heart, skeletal muscle, kidneys, and liver of mice and rats, leading to normalizing plasma BCAA concentrations [17,90,96,97]. A fresh perspective was observed regarding Pp2cm and its interplay with the microbiome metabolite butyrate in the early stages of obesity-related heart failure with preserved ejection fraction (HfpEF) [98]. It has been noted that a decrease in the expression of the Ppm1k gene, responsible for encoding PP2Cm, leads to elevated levels of inactive p-BCKDH, prompting the necessity for additional investigation [14,98,99,100].

Furthermore, BT2 administration in TAC-induced HF delayed progression in mice. These findings suggest that impaired cardiac BCAA catabolism contributes to HF pathogenesis and progression, with BT2 treatment being a possible potent amino acid catabolic restaurateur [66,101,102]. However, the precise impairment in cardiac-specific BCAA dysfunction remains unclear, as these models impact systemic BCAA metabolism, altering both circulating and cardiac BCAA concentrations (Table 2) [56,84,103].

This downregulation is consistent with the disruption of BCAA breakdown. Enhancing BCAA catabolism could be a therapeutic strategy for improving cardiac insulin sensitivity and limiting pathological remodeling after a myocardial infarction [87,108].

After attempting to determine the potential mechanisms underlying fluctuations in plasma and cardiac amino acid concentrations in HF without conclusive results, attention shifted to unraveling the intricate interplay between BCAA delivery to the heart and intrinsic cardiac metabolism in HF pathogenesis regarding mTOR activation [109,110].

Oral supplementation with BCAAs in mice after MI exacerbated cardiac dysfunction by the increased phosphorylation of mTOR, quantified by measuring its hallmark downstream targets, such as ribosomal protein S6 kinase-b1 (also known as P70S6K) and insulin receptor substrate [111], which could be prevented by concurrent treatment with the mTOR inhibitor rapamycin [39,86,112]. Elevated cardiac BCAA concentrations are also linked to mTOR activation in cardiac tissue from humans with dilated cardiomyopathy, associated with the reduced phosphorylation of enzymes involved in insulin signaling, potentially contributing to cardiac insulin resistance via AKT and glycogen synthase kinase 3 [66,99].

The metabolic circadian pattern to which mTOR is subjected could not establish a causal relationship that would explain the daily metabolic fluctuations of BCAAs during heart failure but drew attention to KLF15 [113], a major regulator of the circadian metabolism of BCAAs [114,115]. KLF15, a member of the KLF family, is responsible for activating the mitochondrial branched-chain aminotransferase (BCATm), which is the first step in the breakdown of BCAA [115]. Furthermore, KLF15 can influence the hypertrophic signaling of mTOR in skeletal muscle, causing cardiac insulin resistance and hypertrophy [115]. However, the activity of KLF15 is reduced by the activation of p38 mitogen-activated protein kinase (p38 MAPK), and it is also downregulated in both murine models of heart failure (HF) and patients with cardiomyopathy, leading to BCKA intramyocardial accumulation and BCAA catabolic gene suppression [14,111]. The high concentrations of BCAAs impact mitochondrial ATP production and inhibit pyruvate and 2-KG dehydrogenases, creating a proarrhythmic substrate secondary to impaired contractility, as highlighted in a mice model lacking KLF15 [14]. The information surrounding the regulatory molecules of KLF15 is limited, and previous research has indicated that the activation of p38 MAPK through TGFβ-activated kinase 1 (TAK1) has an inhibitory role by stimulating the expression of transforming growth factor beta (TGFβ) in myocytes, contributing to the impaired BCAA catabolism observed in human DCM hearts (Figure 2) [5,66,105].

There is still more research to be conducted on the topic of myocardial insulin resistance mediated by BCAAs because no clear mechanism has been found. Although research on pancreatic islet cells from mice lacking BCAT2 suggested that BCAAs and their derivatives, BCKAs, would have similar effects, they showed that KIC stimulation of insulin production depends on the conversion of KIC and glutamate to leucine and α-ketoglutarate [111]. Furthermore, mice deficient in BCAT2 showed enhanced insulin resistance and suppressed glucose oxidation in response to BCKA stimulation [111].

### 4.2. Cardiometabolic Disease

Diabetes has emerged as a widespread health crisis, placing a significant strain on healthcare systems globally due to its substantial impact on morbidity and mortality rates. BCAA and their related metabolites are now recognized as highly reliable biomarkers for various cardiometabolic diseases and associated conditions, such as obesity, insulin resistance, and T2D [4]. Several prospective studies have consistently shown a link between circulating levels of BCAAs and the onset of T2DM across different age groups [116].

Current research suggests that BCAAs hold promise in predicting the development of diabetes, with alterations in plasma-free amino acid (AA) profiles observed prior to the onset of T2DM and significantly correlated with future diabetes diagnoses [117]. These changes in plasma-free AA profiles levels likely stem from metabolic shifts triggered by the early stages of diabetes progression. A study analyzing metabolite profiles in 2422 normoglycemic individuals revealed that 201 participants developed diabetes over a 12-year period, highlighting robust associations between the three aforementioned BCAAs, two aromatic AAs (tyrosine and phenylalanine), and diabetic outcomes [117]. These results underscore the potential value of AA profiles in enhancing future risk assessment tools for diabetes.

For instance, a large cohort study involving over 50,000 individuals from the UK Biobank showed that adding plasma nuclear magnetic resonance-measured metabolites, including BCAAs, improved the prediction of T2DM risk when combined with basic clinical parameters [118]. Another prospective cohort study in the Chinese population demonstrated enhanced risk estimation for T2DM prediction by including specific metabolites over an average follow-up period of eight years [119].

A significant study utilizing the Framingham cohort revealed that initial BCAA levels were identified as robust predictors of developing diabetes, with the patients included being three times more likely to develop diabetes within a 12-year monitoring period [77]. This correlation has been consistently verified across various cohorts; however, most studies have primarily examined the risk of developing diabetes based on single-point measurements of circulating BCAA levels [77]. One example is a group of men from South Africa from the SABRE (Southall and Brent Revisited) Study, wherein elevated circulating BCAA levels were linked to around a 40% higher relative risk of incident diabetes [78]. Recently, a longitudinal study in a diverse cohort of young Black and White adults over more than 35 years old indicated that circulating BCAA levels remain stable over a period of 28 years for most young adults, highlighting the necessity of successive metabolomic assessments to identify specific subgroups exhibiting an elevated risk of developing diabetes later in life [120].

Additionally, an updated meta-analysis incorporating data from 61 prospective cohort reports on T2DM risk underscored strong associations not only with BCAAs but also other amino acids and lipid metabolites [121]. This analysis evaluated metabolite concentrations in various biological samples, such as plasma, serum, and urine samples, using high-throughput metabolomic platforms [121].

Also elevated levels of 3-HIB (3- hydroxyisobutyrate) in plasma serve as an indicator for future T2DM risk and may play a crucial role in regulating metabolic flexibility in the heart and muscles [12,117]. While diabetic conditions lead to increased circulating levels of branched-chain amino acids (BCAA), ketones, glucose, and fatty acids, their oxidation rates do not proportionately increase this imbalance, contributing to cardiac dysfunction among diabetic individuals [12].

A meta-analysis of 19 prospective studies revealed an association between branched-chain amino acids (BCAAs) and aromatic AAs with prediabetes and T2DM, which are mechanistically related through their potential impact on insulin resistance [122]. The activation of mTOR/p70S6 kinase by increased BCAAs, particularly leucine, has been proposed as the most prevalent mechanism that inhibits insulin signaling by phosphorylating the IRS-1 [32]. However, it remains unclear whether BCAA-induced mTORC1 activation is sufficient to trigger insulin resistance or if further research is needed to understand their roles fully [32].

Recently, attention has been directed towards trimethylamine-N-oxide (TMAO), a metabolite produced as a result of gut microbiota fermentation that has garnered attention due to its suspected involvement in the progression of ischemic heart disease [123], complications associated with T2DM, and premature mortality in the general population [124,125]. While some studies have linked elevated plasma levels of TMAO with adverse cardiovascular outcomes, findings among individuals with underlying health conditions vary [126]. Notably, the association between TMAO and cardiovascular mortality in individuals with T2DM seems to be particularly significant among high-risk populations [126]. Recent research has indicated that BCAA concentrations are also influenced by the microbiota, therefore raising the question of how circulating BCAA concentrations correlate with TMAO levels in diabetic milieu [126,127].

Furthermore, research suggests that excessive BCAA catabolism may play a role in the pathway from obesity and insulin resistance to T2DM by influencing inflammation and genetically predicted insulin resistance [128].

### 4.3. Hypertension

The prevalence of high blood pressure and the negative effects it has on health, as well as the associated economic burdens, are well known risk factors for CVDs [31,129,130,131]. While studies on heart failure and coronary artery disease are more frequent and several pathophysiological mechanisms have been identified regarding hypertension, the results are inconclusive, pointing only to an association between BCAAs and this condition [31,129,130,131].

Since Akt/mTORC1 was highlighted in connection to heart failure, researchers aimed to uncover its role in hypertension [132,133]. Potential interplay between BCAAs and the increased activation of angiotensin-converting enzymes through binding amino acid residues has been proposed [132,133]. A study published by Chen et al. reinforced this supposition by uncovering elevated levels of mTOR in hypertensive rats with cardiac hypertrophy [134]. Consequently, the mTOR/S6K1 pathway was found to be activated by angiotensin II (ANG II), further promoting oxidative stress and decreased levels of nitric oxide (NO) [134].

While the particular metabolic pathways remain under scrutiny, a few studies have tried to underline a correlation between plasma BCAA levels and dietary intake with hypertension [135]. Although attempts have sought to determine precisely which amino acid is linked to hypertension risk, studies have yielded conflicting results [135]. For example, in an Iranian study involving 4288 non-hypertensive patients, dietary Val intake turned out to have a stronger correlation with hypertension over a monitoring period of three years [136,137], while Mahbub et al. discovered the involvement of Leu in Japanese participants [132].

When measuring 24 h urine levels of BCAAs in older hypertensive patients, isoleucine was linked with diastolic blood pressure, while valine was found to be negatively correlated with both systolic and diastolic blood pressure [138]. However, Siomkajło et al. [131] found significant positive associations between plasma levels and both systolic and diastolic blood pressure findings reinforced by a large-scale prospective cohort study on middle-aged men and women conducted by Flores-Guerrero et al. [139]. On the other hand, Liu et al. found no particular association between a specific BCAA with hypertensive risk, thus raising the hypothesis of population-based variability and genetic background [135].

The PREVEND study, enrolling over 4100 participants over nine years of follow-up, predicted the impact of incident hypertension on BCAA plasma levels independent of traditional risk factors [139]. Similar results were observed in cross-sectional analyses of two Asian cohorts (81 out of 8589 Japanese individuals and 82 out of 472 Chinese individuals) [140]. On the other hand, a lower incidence of arterial stiffness and incident hypertension was linked to a higher dietary intake of BCAAs in research involving 1898 female twins from the TwinsUK registry, with the caveat that plasma levels were also influenced by intrinsic BCAA metabolic impairment, not only by dietary consumption [130].

The precise mechanisms underlying the connection between each BCAA and hypertension have yet to be clarified. Regarding leucine, a partial mechanism responsible for a stronger association with high blood pressure was proposed via the inhibition of NO synthesis from L-arginine and alanine generation secondary to glutamate production after the transamination process [132,136].

While the aforementioned studies found discrepant results, it has been consistently shown that BCAA concentrations are positively associated with high blood pressure, with the reserve that other nutrients or yet unknown pathways may counterbalance their effects (Table 3) [141].

### 4.4. Atherosclerosis and Coronary Artery Disease

Peripheral artery disease, ischemic stroke, and atherosclerotic coronary heart disease are only a few of the major cardiovascular events that are caused by atherosclerosis (AS), and the significantly increase the morbidity and mortality rates associated with CVD [144,145]. Besides the traditional risk factors extensively researched in AS pathogenesis, BCAAs have also sparked interest; however, the exact mechanisms are still unknown [1,146,147].

Two of the initial steps in the development of atherosclerotic plaque are represented by lipid and cholesterol deposition at the endothelial level, followed by inflammatory cell infiltration, which facilitates platelet activation and thrombus formation after a plaque rupture [148]. A comparative study between human platelets and platelets from Ppm1k^−^/^−^ mice showed that BCAA consumption in human volunteers led to platelet activation and aggregation because of collagen [149]. On the other hand, platelets from Ppm1k^−^/^−^ mice exhibited lower susceptibility to activation and thrombus formation acting indirectly upon the thrombotic process via intermediate catabolic products, especially α-ketoisovalerate [149]. Another aspect worth mentioning is that Ppm1k^−^/^−^ mice with increased BCAA plasma levels had an inauspicious response to ischemia–reperfusion injury at the expense of downregulated glucose oxidation [1,28,150].

Zhao et al. discovered that increased BCAA levels lead to proinflammatory macrophage activation and AS progression [151]. Mitochondrial H2O2 (mtH2O2) levels observed in activated macrophages induced the formation of high-mobility group box 1 protein (HMGB1) and the activation of the Toll-like receptor 4 (TLR4)/nuclear factor kappa-light-chain-enhancer of activate B cells (NF-κB) pathway, which is modulated by BCAA catabolism [151]. Notably, recent research by Zhenyukh et al. has suggested that elevated levels of BCAAs stimulate the PI3K-Akt/mTORC1 signaling pathway, triggering the formation of reactive oxygen species (ROS) and proinflammatory proteins [38,152,153]. Nevertheless, the underlying molecular pathways are yet to be understood, and it is unclear whether an increased expression of BCAAs is linked to macrophage-mediated atherosclerotic inflammation [35,38,154].

mTOR activation was also linked to reduced endothelial relaxation secondary to leucine levels in mouse aortic rings, in which an increased phosphorylation of 40S ribosomal protein S6 and overexpression of ribosomal protein S6 kinase (S6K) were observed (Figure 2) [155].

Concerning coronary artery disease, BCAAs have highlighted themselves as potential biomarkers and a severity predictor independently or in association with T2DM [144,156,157]. Being one of the primary causes of cardiovascular death, the coexistence with T2DM increases the severity of lesions, worsening the prognosis and reducing patient survival due to severe complications [1].

Baseline plasma levels of BCAAs and the risk of acute HF complication in patients with ST segment elevation MI (STEMI) necessitating percutaneous coronary intervention were first observed by Shah et al. in 2012 [158] and were associated with a worse prognosis [86,109]. Later, the PREDIMED study concluded that an elevated risk of myocardial infarction was associated with leucine and isoleucine levels [64], validating earlier findings from a European cohort in which isoleucine was linked to atherosclerosis progression (Table 4) [70,159].

A Women’s Health Study that included over 27,000 participants with 19 years of follow-up reinforced the predictive role of BCAA levels in identifying patients that developed CVD in association with T2DM [69,164]. A potential limitation was based on race since the analysis from the Diabetes Heart Study showed a difference between European Americans and African Americans in BCAA levels, which showed a correlation with coronary artery calcium [165]. Another particularity in African Americans was observed in the Jackson Heart Study, where leucine was associated with a lower incidence of coronary artery disease [166].

## 5. Discussion

The molecular ties between BCAAs and chronic CVDs represent a complex and evolving area of research with important implications for disease pathogenesis and therapeutic strategies. Throughout this exploration, several key insights and conclusions can be drawn.

The balance of free circulating BCAAs is influenced by both their intake (from diet and proteolysis) and their utilization (for protein synthesis and catabolism for energy). As essential amino acids that can be obtained only from food sources, their homeostasis must be maintained by catabolism [1,8]. The first two steps of this process are the same for all three BCAAs: BCAT catalyzes reversible transamination, which forms BCKAs. The irreversible oxidative decarboxylation of BCKAs, which is subsequently catalyzed by the BCKDH complex, represents the rate-limiting step in the entire BCAA catabolic complex process (Figure 1) [1,8].

Elevated circulating BCAA levels are commonly observed in individuals with chronic CVDs, including heart failure, atherosclerosis, and hypertension. This dysregulation of BCAA metabolism and accumulation in the bloodstream are closely linked to insulin resistance, dyslipidemia, oxidative stress, and endothelial dysfunction, which are critical factors in CVD development and progression [65,70]. Although they are crucial in numerous essential processes, their deleterious effect is secondary to imbalances occurring in one of the phases of the complex metabolism, thus altering their plasmatic level secondary to local cardiac-level fluctuations in different pathologic states [65,70]. The molecular links between BCAAs and CVD can be understood through several mechanisms.

Elevated BCAA levels are associated with insulin resistance, a condition that often precedes type 2 diabetes [48]. Insulin resistance itself is a significant risk factor for cardiovascular diseases [48]. BCAAs may impair insulin signaling pathways, particularly through the mTOR pathway and serine phosphorylation of insulin receptor substrate (IRS) [11,26,32]. Chronic activation of the mTOR pathway desensitizes cells to insulin, contributing to insulin resistance, especially secondary to leucine [2,47]. BCAAs can also promote the phosphorylation of IRS on serine residues rather than tyrosine residues, disrupting the insulin signaling cascade and further sustaining insulin resistance [11,26,32]. Dysregulated mTOR signaling, influenced by BCAA availability, can contribute to adverse cardiac remodeling, fibrosis, and hypertrophy, ultimately impacting cardiovascular function. BCAAs interact with transcriptional regulators like SREBPs and PPARs, affecting lipid metabolism and inflammatory processes in the cardiovascular system [1,50,51,52]. Inflammatory pathways such as the NF-κB pathway, which is known to contribute to atherosclerosis, are upregulated by elevated levels of BCAAs [151]. This can also lead to oxidative stress by altering mitochondrial function and promoting the production of reactive oxygen species (ROS). Oxidative stress damages the vascular endothelium, contributing to atherosclerosis and other cardiovascular issues [38,152,153].

Regarding dyslipidemia, a known risk factor for cardiovascular diseases, BCAAs modulate lipid metabolism by affecting the expression of genes involved in lipid synthesis and degradation, leading to increased levels of circulating free fatty acids, triglycerides, and low-density lipoprotein (LDL) cholesterol, promoting atherosclerosis and coronary artery disease [167].

Besides their association with the traditional cardiovascular risk factors, gut microbiota play a significant role in metabolizing BCAAs [126]. Dysbiosis, or an imbalance in gut microbiota, can lead to altered BCAA metabolism and increased levels of harmful metabolites such as TMAO, which has been linked to atherosclerosis and CVDs [126].

In heart failure, there is a decrease in the central circadian regulator of BCAA metabolism, KLF15. This downregulation is mediated by the TAK1 and p38 MAPK signaling pathway, resulting in a reduced expression of BCAA metabolic enzymes such as mitochondrial BCAT2, BCKDH, and Ppm1k [5,66,105]. Consequently, there is an accumulation of BCAAs and BCKAs at cardiac level, while cardiomyocyte injury may also disrupt BCAA metabolism in peripheral tissues, leading to elevated circulating levels of BCAAs and BCKAs [111]. As already mentioned, Leu plays a role in activating mTOR, inhibiting autophagy through ULK1, inducing insulin resistance via IRS-1 phosphorylation by S6K, and promoting protein synthesis through 4E-BP1 phosphorylation. BCKAs contribute to increased 4E-BP1 phosphorylation and activate the MEK-ERK pathway [11,26,32]. Exposure to BCKAs impairs mitochondrial complex I function in the heart, causing oxidative stress due to superoxide generation [11,26,32].

During ischemia–reperfusion injury, deleting the Ppm1k gene in mice leads to elevated levels of both circulating and cardiac BCAAs and BCKAs [1,28,150]. This increase hinders glucose transport and oxidation by reducing the O-linked N-acetylglucosaminylation of pyruvate dehydrogenase (PDH), exacerbating ischemic damage [1,28,150]. Through an mTOR-dependent mechanism involving ROS production, BCAAs interfere with vascular relaxation [155]. Platelet oxidation of BCAAs promotes thrombosis by propionylating tropomodulin 3, a potent platelet activator [35,38,154].

In models of obesity and insulin resistance in animals, accumulated BCAAs hinder fatty acid oxidation while boosting triglyceride storage. Elevated levels of BCKAs seen in obesity and type 2 diabetes inhibit AKT and PDH, affecting energy substrate utilization, but the impact of these alterations on cardiac function remains uncertain at present [28,150]. It is also unclear whether changes within the cardiac muscle trigger shifts in BCAA metabolism within other tissues like the liver, skeletal muscle, or adipose tissue, and this is a question that persists regarding TMAO also [123].

Numerous human epidemiological studies have indicated a correlation between circulating BCAAs and the onset of cardiometabolic diseases [120]. However, most of these studies relied on single-time measurements of BCAAs in populations of European ancestry, and fasting levels of BCAAs exhibit significant variations between men and women, suggesting potential sex-specific differences in how BCAAs relate to the risk of cardiometabolic diseases [77]. This complexity underscores the importance of quantifying the associated risks more effectively by utilizing repeated measures and exploring these connections across diverse cohorts consisting of both sexes and different races [126]. Adopting such an approach will not only help in assessing the dynamic nature of circulating BCAAs but also in identifying the factors influencing the relationships between BCAAs and specific cardiometabolic conditions at molecular levels.

## 6. Conclusions

In summary, the role of BCAAs in chronic CVD underscores their multifaceted involvement in metabolic regulation, cellular signaling, and disease pathophysiology [168]. Despite significant advances, many questions remain regarding the precise mechanisms underlying BCAA-mediated effects on cardiovascular health.

By unraveling the complexities of BCAA metabolism and its interactions with cardiovascular health, researchers can identify novel therapeutic targets and dietary interventions aimed at improving outcomes for individuals with chronic cardiovascular conditions. However, it is essential to note that many additional factors, such as overall diet, genetics, and lifestyle, influence individual risk profile. Continued investigation into the role of BCAAs in CVDs holds promise for advancing our understanding of disease mechanisms and developing more effective strategies for prevention and treatment.

## Figures and Tables

**Figure 1 nutrients-16-01972-f001:**
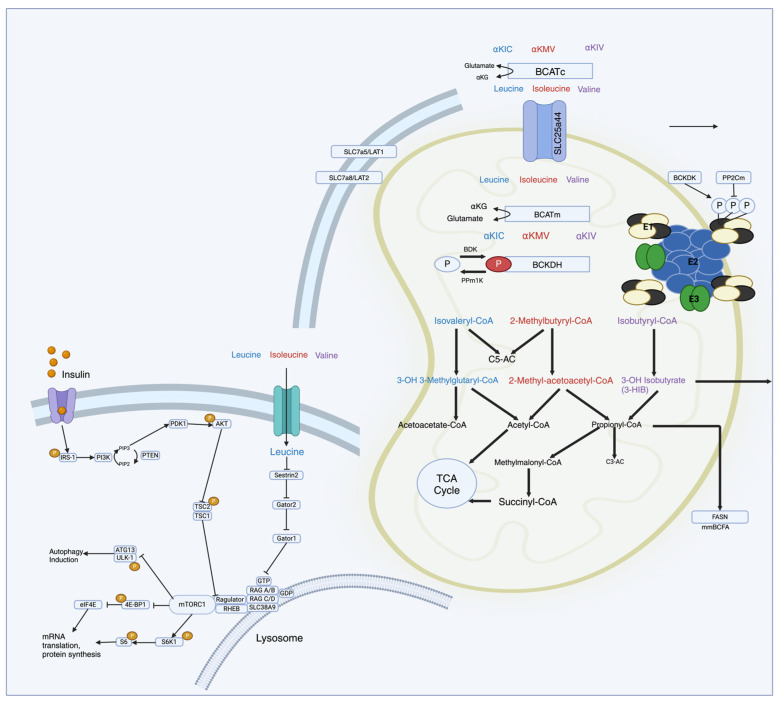
Schematic representation of molecular pathways of BCAA metabolism in relation to mTOR regulation. Upon ingestion, BCAAs are transported into cells and undergo metabolism. The first step involves transamination, where BCAAs are converted into BCKAs by the enzyme BCAT. Subsequently, the BCKAs are further metabolized in mitochondria through a series of enzymatic reactions, ultimately generating intermediates like acetyl-CoA and succinyl-CoA, which enter central metabolic pathways. Leu acts as a direct activator of mTORC1 that stimulates protein synthesis by phosphorylating downstream effectors like S6 kinase 1 (S6K1) and eukaryotic translation initiation factor 4E-binding protein 1 (4E-BP1), leading to the enhanced translation of messenger ribonucleic acid (mRNA) into proteins involved in cell growth and proliferation.

**Figure 2 nutrients-16-01972-f002:**
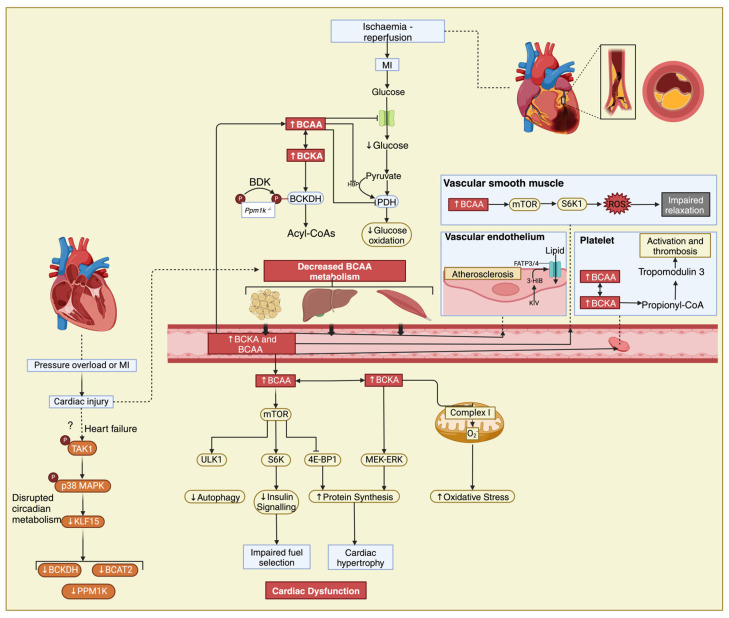
The interplay between BCAAs, heart failure, and atherosclerosis. Elevated circulating levels of BCAAs have been observed in individuals with heart failure and have been linked to insulin resistance, which is a common feature of both heart failure and atherosclerosis. In the setting of myocardial infarction, disrupted BCAA metabolism can impact mitochondrial function in cardiac muscle cells, potentially exacerbating tissue damage and impairing recovery post-MI. ?—Cardiac injury also potentially impairs BCAA metabolism via a mechanism involving transforming growth factor-ß-activated kinase 1 (TAK1) and p38 mitogen-activated protein kinase (MAPK) signalling and downregulation of KLF15 as seen in heart failure, resulting in increased circulating BCAA and BCKA levels. Additionally, BCAAs may play a role in promoting inflammation and oxidative stress, contributing to the progression of atherosclerosis. Unv-51-like kinase 1, ULK1; ribosomal protein S6 kinase, S6K; 4E-binding protein 1, 4E-BP1; mitogen-activated protein kinase ERK kinase/extracellular-signal-regulated kinase, MEK-ERK.

**Table 1 nutrients-16-01972-t001:** Summary of references for studies on BCAAs as potential biomarkers.

Study Design	Results	Ref.
Case–cohort of 970 patients PREDIMED trial	70% excess risk of CVD and stroke when associated with high levels of BCAA	
Cohort	Increased BCAA levels in diabetic cardiomyopathy secondary to TAK1/P38 MAPK axis KLF15 inhibition	[66]
Cohort	BCAAs associated with cardiovascular mortality	[67]
Randomized double- blinded placebo- controlled crossover study	Leucine does not influence insulin resistance	[68]
Prospective cohort of U.S. Women’s Health Study including 27,041 women	BCAA levels associated with incident CVD in women	[69]
Case–control study from Malmö Diet and Cancer Cardiovascular Cohort (MDC-CC)	BCAA levels predict CVD	[70]
Case–cohort study from FINRISK	BCAA levels as a CVD risk factor	[71]
Cohort from ADVANCED study	BCAA levels associated with major macrovascular complications in T2D	[72]
Cohort from Malmö Diet and Cancer Cardiovascular Cohort (MDC-CC)	Valine and isoleucine associated with an increased CVD risk	[73]
Cohort from Concord Health and Ageing in Men Project (CHAMP)	Lower BCAA levels associated with higher mortality and major cardiovascular endpoints (MACEs)	[74]
685 participants without diabetes of the Insulin Resistance Atherosclerosis Study (IRAS)	BCAA levels associated with insulin resistance and T2D	[75]
Genome-wide study of 16,596 patients	BCAA levels associated with a higher risk of T2D	[76]
Case–control study of 2422 patients in the Framingham cohort	BCAA levels associated with a higher risk of T2D	[77]
Cohort of 1279 European and 1007 South Asian patients	BCAA levels associated with a 40% higher risk of T2D	[78]

TGF-β activated kinase-1/p38 mitogen-activated protein kinase (TAK1/P38MAPK); Krüppel-like factor 15 (KLF15); major adverse cardiovascular events (MACE).

**Table 2 nutrients-16-01972-t002:** Summary of references for relevant studies related to BCAAs and heart failure.

Study Design	Results	Ref.
Experimental study (pigs)	Empagliflozin ameliorates left ventricular systolic function via BCAA myocardial consumption	[80]
Experimental study (Dahl salt-sensitive rats fed high-salt diet)	BCAA prolonged survival in HF	[104]
Experimental study (PP2Cm-knockout mice)	BCAAs impaired myocardial contractions	[90]
Experimental study (murine model)	BCAAs associated with post-MI HF	[86]
Prospective observational study of 29,103 patients	BCAAs levels associated with HF in T2D	[105]
Crossover controlled trial	BCAA supplementation improved serum albumin levels in HF	[106]
Randomized controlled study including 1032 patients	Leucine and valine associated with higher mortality in HF	[107]
Prospective study	BCAA associated with diastolic left ventricular function	[58]

**Table 3 nutrients-16-01972-t003:** Summary of references on relevant studies related to BCAAs and hypertension.

Study Design	Results	Ref.
Prospective cohort of 4288 participants	BCAA intake (valine) associated with a higher incidence of hypertension	[137]
Cross-sectional study	Amino acid intake increases peripheral blood pressure	[142]
Cross-sectional study	BCAA intake associated with a higher incidence of hypertension	[143]

**Table 4 nutrients-16-01972-t004:** Summary of references for relevant studies related to BCAAs, atherosclerosis, and coronary artery disease.

Study Design	Results	Ref.
Experimental study	BCAA levels associated with AS pathogenesis	[151]
Cohort	BCAA levels associated with coronary and carotid atherosclerosis	[160]
Experimental study (902 patients)	Increased BCAA levels associated with cardiovascular events in patients with STEMI and acute HF	[110]
Case–control study population of 1983 patients undergoing coronary angiography	BCAA independently associated with CAD diagnosis	[161]
Case–cohort, prospective, population based	BCAA levels associated with CAD	[162]
Case–control	BCAA levels associated with CAD	[163]
Experimental study (adult mice)	BCAA enhances I/R injury via CN2/ATF6/PPAR-α pathway	[53]
Experimental study (Wild-type C57BL/6 mice)	Overexpression of PP2Cm alleviates MI/R injury by reducing BCAA catabolic impairment	[28]
Experimental study (Male C57BL/6 mice and Wistar rats)	PI3K/Akt/GSK3β pathway attenuates myocardial I/R injury	[38]

Coronary artery disease, CAD; protein activating transcription factor 6, ATF6; peroxisome proliferator-activated receptor, PPAR; phosphatidylinositol 3-kinase/serine/threonine protein kinase/glycogen synthase kinase 3, PI3K/Akt/GSK3β; ischemia–reperfusion, I/R.

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
