# Peer review of "Duality of Branched-Chain Amino Acids in Chronic Cardiovascular Disease: Potential Biomarkers versus Active Pathophysiological Promoters"

_nutrients, 2024, doi:10.3390/nu16121972_

Round 1
Reviewer 1 Report
Comments and Suggestions for Authors
The narrative review of Tanase and colleagues explores the role of amino acids and in particular branched-chain amino acids in cardiovascular disease. The review is overall well written and organized. Please find below specific comments:
- I would reorganize the abstract expanding the section on specific review aims instead of writing a long background section.
- Please cite the appropriate references when needed (e.g. the following statement is not supported by any reference "Alterations in metabolism and circulating BCAA levels have been observed and are 65 increasingly recognized as potential biomarkers and are associated with insulin re-66 sistance, dyslipidemia, and endothelial dysfunction").
- Diabetes should be an important sub-paragraph, before hypertension. There are several studies on AAs and BCAAs and diabetes. Both circulating levels and aminoaciduria have shown to play important role in both type 1 and type 2 diabetes and related complications such as diabetic ketoacidosis. To this regard, in addition to animal models (the cited ref. 51) there are human studies demonstrating the importance of AAs for the diagnosis and characterization of patients with diabetes (see doi: 10.1007/s00467-022-05531-3)
Author Response
Dear Reviewer,
Firstly, we want to thank you on behalf of our team for your time and all your observations regarding our manuscript. As recommended, we have revised the abstract, and completed for a better overview of its content. To further improve this manuscript, we have corrected any grammatical errors identified through the paper, rephrased when necessary and improved structural issues. We have taken in consideration all your advices and made the following changes:
- I would reorganize the abstract expanding the section on specific review aims instead of writing a long background section.
We followed your recommendation rephrased and introduced a new paragraph in the abstract in which we described concisely the objectives of our review.
- Please cite the appropriate references when needed (e.g. the following statement is not supported by any reference "Alterations in metabolism and circulating BCAA levels have been observed and are 65 increasingly recognized as potential biomarkers and are associated with insulin re-66 sistance, dyslipidemia, and endothelial dysfunction").
Thank you for this observation. Also, we identified as much as possible all the missing references, pointed and corrected them.
- Diabetes should be an important sub-paragraph, before hypertension. There are several studies on AAs and BCAAs and diabetes. Both circulating levels and aminoaciduria have shown to play important role in both type 1 and type 2 diabetes and related complications such as diabetic ketoacidosis. To this regard, in addition to animal models (the cited ref. 51) there are human studies demonstrating the importance of AAs for the diagnosis and characterization of patients with diabetes (see doi: 10.1007/s00467-022-05531-3)
Thank you for this observation. The concept of diabetes and BCAAs interplay was briefly included in the section “4. Branched-chain amino acid in cardiovascular disease” (line 489-491, line 677-681, line 695-697 in correlation to each cardiovascular disease depicted.
We however, taken into consideration your valuable comments and we have accordingly included a paragraph with additional information about diabetes, with relevance to our theme and added the corresponding references ; as we wanted the review to be exclusively dedicated to cardiovascular pathology in order to keep the continuity of the manuscript we added the paragraph before “4.1. Heart failure”.
“ A significant study utilizing the Framingham cohort revealed that initial BCAA levels were identified as robust predictors of developing diabetes, patients included being three times more likely to develop diabetes within a 12-year monitoring period [77]. This correlation has been consistently verified across various cohorts, however most studies have primarily examined the risk of developing diabetes based on single-point measurements of circulating BCAA levels during middle age [77]. One example is a group of men from South Africa from the SABRE (Southall and Brent Revisited) Study, where elevated circulating BCAA levels were linked to around a 40% higher relative risk of incident diabetes [78]. Recently, a longitudinal study in a diverse cohort of young Black and White adults over more than 35 years indicated that circulating BCAA levels remain stable over a period of 28 years for most young adults, highlithing the necesity of successive metabolomic assessments to identify specific subgroups exhibiting an elevated risk of developing diabetes later in life [82].
Recent attention is directed towards trimethylamine-N-oxide (TMAO), a metabolite produced as a result of gut microbiota fermentation that has garnered attention due to its suspected involvement in the progression of ischemic heart disease [83], complications associated with T2DM, and premature mortality in the general population [84,85].
While some studies have linked elevated plasma levels of TMAO with adverse cardiovascular outcomes, findings among individuals with underlying health conditions vary [86]. Notably, the association between TMAO and cardiovascular mortality in individuals with T2DM seems to be particularly significant among high-risk populations [86]. Recent research has indicated that BCAA concentrations are also influenced by the microbiota therefore, raising the question how circulating BCAA concentrations correlate with TMAO levels in diabetic milieu [86,87].”
- Wang, T.J.; Larson, M.G.; Vasan, R.S.; Cheng, S.; Rhee, E.P.; McCabe, E.; Lewis, G.D.; Fox, C.S.; Jacques, P.F.; Fernandez, C.; et al. Metabolite Profiles and the Risk of Developing Diabetes. Nat Med 2011, 17, 448–453, doi:10.1038/nm.2307.
- Tillin, T.; Hughes, A.D.; Wang, Q.; Würtz, P.; Ala-Korpela, M.; Sattar, N.; Forouhi, N.G.; Godsland, I.F.; Eastwood, S.V.; McKeigue, P.M.; et al. Diabetes Risk and Amino Acid Profiles: Cross-Sectional and Prospective Analyses of Ethnicity, Amino Acids and Diabetes in a South Asian and European Cohort from the SABRE (Southall And Brent REvisited) Study. Diabetologia 2015, 58, 968–979, doi:10.1007/s00125-015-3517-8.
- Sawicki, K.T.; Ning, H.; Allen, N.B.; Carnethon, M.R.; Wallia, A.; Otvos, J.D.; Ben-Sahra, I.; McNally, E.M.; Snell-Bergeon, J.K.; Wilkins, J.T. Longitudinal Trajectories of Branched Chain Amino Acids through Young Adulthood and Diabetes in Later Life. JCI Insight 2023, 8, e166956, doi:10.1172/jci.insight.166956.
- Gencer, B.; Li, X.S.; Gurmu, Y.; Bonaca, M.P.; Morrow, D.A.; Cohen, M.; Bhatt, D.L.; Steg, P.G.; Storey, R.F.; Johanson, P.; et al. Gut Microbiota‐Dependent Trimethylamine N‐oxide and Cardiovascular Outcomes in Patients With Prior Myocardial Infarction: A Nested Case Control Study From the PEGASUS‐TIMI 54 Trial. JAHA 2020, 9, e015331, doi:10.1161/JAHA.119.015331.
- Croyal, M.; Saulnier, P.-J.; Aguesse, A.; Gand, E.; Ragot, S.; Roussel, R.; Halimi, J.-M.; Ducrocq, G.; Cariou, B.; Montaigne, D.; et al. Plasma Trimethylamine N-Oxide and Risk of Cardiovascular Events in Patients With Type 2 Diabetes. The Journal of Clinical Endocrinology & Metabolism 2020, 105, 2371–2380, doi:10.1210/clinem/dgaa188.
- Cardona, A.; O’Brien, A.; Bernier, M.C.; Somogyi, A.; Wysocki, V.H.; Smart, S.; He, X.; Ambrosio, G.; Hsueh, W.A.; Raman, S.V. Trimethylamine N-Oxide and Incident Atherosclerotic Events in High-Risk Individuals with Diabetes: An ACCORD Trial Post Hoc Analysis. BMJ Open Diab Res Care 2019, 7, e000718, doi:10.1136/bmjdrc-2019-000718.
- Tang, W.H.W.; Wang, Z.; Li, X.S.; Fan, Y.; Li, D.S.; Wu, Y.; Hazen, S.L. Increased Trimethylamine N-Oxide Portends High Mortality Risk Independent of Glycemic Control in Patients with Type 2 Diabetes Mellitus. Clinical Chemistry 2017, 63, 297–306, doi:10.1373/clinchem.2016.263640.
- Flores-Guerrero, J.L.; Van Dijk, P.R.; Connelly, M.A.; Garcia, E.; Bilo, H.J.G.; Navis, G.; Bakker, S.J.L.; Dullaart, R.P.F. Circulating Trimethylamine N-Oxide Is Associated with Increased Risk of Cardiovascular Mortality in Type-2 Diabetes: Results from a Dutch Diabetes Cohort (ZODIAC-59). JCM 2021, 10, 2269, doi:10.3390/jcm10112269.
Thank you once again for your all your suggestions, we tried to revise, correct, restructure and correct any mistakes, and ultimately improve our paper.

Reviewer 2 Report
Comments and Suggestions for Authors
This is a very comprehensive review of the molecular regulation of BCAAs. Although extremely thorough, the message of the review is a bit lost in the narrative of how BCAAs are regulated. It reads more like a lecture on BCAA regulation and less like a link to potential ties to CVD. Additionally, since BCAAs are essential, it is paradoxical that they would be detrimental to cardiovascular health, and this aspect has not been rationalized in this manuscript. With that said, please see my comments below that in my opinion would strengthen this manuscript:
1. Please ensure that once you introduce an acronym, it is used consistently throughout the rest of the text. For example, CVD, BCAAs and KLF have been fully spelled out several times throughout. This just ensures cohesion throughout the manuscript.
2. Figure 2 is flipped across the vertical axis ( i.e from right to left ) and thus is fully illegible
3. After reading all the information on how BCAAs are catabolized, I still have a problem reconciling how circulating levels of BCAAs could be instrumental in precipitating CVD. The data is presented that is both pro and con that BCAA levels have anything to do with different CVD, thus leaving the reader a bit perplexed as to what the stance of this review is. I am wondering whether the authors could work on the parts that provide molecular ties to BCAAs being linked to CVD such that this message comes through more clearly.
4. I would argue for a change of the title as well, the word "portrayal" is not suitable in my opinion.
Author Response
Dear Reviewer,
Firstly, thank you on behalf of our team for your time and expertise in peer-reviewing our manuscript. We have taken into consideration all your suggestion and comments, made structural changes, revised the whole manuscript for abbreviations and corrected. Thus, we believe that with the help of your comments we improved the quality of this paper.
- Please ensure that once you introduce an acronym, it is used consistently throughout the rest of the text. For example, CVD, BCAAs and KLF have been fully spelled out several times throughout. This just ensures cohesion throughout the manuscript.
Thank you for this observation, we identified as much as possible all the missing abbreviations, pointed and corrected them.
- Figure 2 is flipped across the vertical axis (i.e from right to left) and thus is fully illegible
Thank you for pointing this out, we really hope we managed to adjust the figure to look clear and be easy for the readers to understand the depicted information.
- After reading all the information on how BCAAs are catabolized, I still have a problem reconciling how circulating levels of BCAAs could be instrumental in precipitating CVD. The data is presented that is both pro and con that BCAA levels have anything to do with different CVD, thus leaving the reader a bit perplexed as to what the stance of this review is. I am wondering whether the authors could work on the parts that provide molecular ties to BCAAs being linked to CVD such that this message comes through more clearly.
Upon revision of the manuscript, we see the point of view of your expert insights therefore to answer to these questions, we have enriched our manuscript with chapter “5.Discussion”. Being a long and comprehensive topic, we tried to summarize the current filed quality recent reviews and focus on the differences behind BCAAs common ground as a biomarker as well as an active mediator of cardiovascular pathology.
- I would argue for a change of the title as well, the word "portrayal" is not suitable in my opinion.
Indeed, we agree it was an improper use of the term, therefore we have replaced it, in order to clarify our content.
Thank you once again for all your thoughtful comments and careful reading we tried to revise, restructure and correct any mistakes and/or grammatical issues, and ultimately improve our paper.

Round 2
Reviewer 1 Report
Comments and Suggestions for Authors
Diabetes is the major cardiovascular risk factor together with hypertension, thus it is part of cardiovascular pathology. Indeed, all cardiovascular risk factors and CVD are associated with systemic endothelial dysfunction and potential biomarkers should be explored in the entire spectrum of conditions.
I recommend to include all cardiovascular risk factors and CVD into consideration in your review to be consistent and complete.
If Authors are not willing to include an entire paragraph on diabetes at least a subparagraph should be included. Furthermore, they stated that most studies on AA and diabetes focused on middle age. However, this is inaccurate since multiple studies on young people have been published so far.
Author Response
Dear Reviewer,
Firstly, we want to thank you on behalf of our team for your time on peer-reviewing our manuscript. Upon revision of the manuscript, indeed we agree diabetes is an important part interconnected with the subject of our paper. Therefore, as recommended to confer unity and to make our paper complete, we have introduced a new subparagraph entitled “4.2. Cardiometabolic disease” in which we briefly describe the association between BCAAs and diabetes and deleted the misleading information which incorrectly stated that most of the studies included only middle aged subjects.
Thank you once again for all your thoughtful suggestions and expert insights.
